# $\mathcal{G}$-SGD: Optimizing ReLU Neural Networks in its Positively Scale-Invariant Space

**Qi Meng**[1][*], **Shuxin Zheng**[2][*][†], **Huishuai Zhang**[1], **Wei Chen**[1], **Qiwei Ye**[1], **Zhi-Ming Ma**[4],
**Nenghai Yu**[3], **Tie-Yan Liu**[1]
[1]Microsoft Research Asia, [2,3]University of Science and Technology of China
[4]University of Chinese Academy of Sciences
[1]{meq, huishuai.zhang, wche, qiwye,tie-yan.liu}@microsoft.com
[2]zhengsx@mail.ustc.edu.cn, [3]ynh@ustc.edu.cn, [4]mazm@amt.ac.cn

## Abstract

It is well known that neural networks with rectified linear units (ReLU) activation functions are positively scale-invariant. Conventional algorithms like stochastic gradient descent optimize the neural networks in the vector space of weights, which is, however, not positively scale-invariant. This mismatch may lead to problems during the optimization process. Then, a natural question is: *can we construct a new vector space that is positively scale-invariant and sufficient to represent ReLU neural networks so as to better facilitate the optimization process* ? In this paper, we provide our positive answer to this question. First, we conduct a formal study on the positive scaling operators which forms a transformation group, denoted as $\mathcal{G}$. We show that the value of a path (i.e. the product of the weights along the path) in the neural network is invariant to positive scaling and prove that the value vector of all the paths is sufficient to represent the neural networks under mild conditions. Second, we show that one can identify some basis paths out of all the paths and prove that the linear span of their value vectors (denoted as $\mathcal{G}$-space) is an invariant space with lower dimension under the positive scaling group. Finally, we design stochastic gradient descent algorithm in $\mathcal{G}$-space (abbreviated as $\mathcal{G}$-SGD) to optimize the value vector of the basis paths of neural networks with little extra cost by leveraging back-propagation. Our experiments show that $\mathcal{G}$-SGD significantly outperforms the conventional SGD algorithm in optimizing ReLU networks on benchmark datasets.

## 1 Introduction

Over the past ten years, neural networks with rectified linear hidden units (ReLU) (Hahnloser et al., 2000) as activation functions have demonstrated the power in many important applications, such as information system (Cheng et al., 2016; Wang et al., 2017), image classification (He et al., 2016a; Huang et al., 2017), text understanding (Vaswani et al., 2017), etc. These networks are usually trained with *Stochastic Gradient Descent* (SGD), where the gradient of loss function with respect to the weights can be efficiently computed via back propagation method (Rumelhart et al., 1986).

Recent studies (Neyshabur et al., 2015; LeCun et al., 2015) show that ReLU networks have positively scale-invariant property, i.e., if the incoming weights of a hidden node with ReLU activation are multiplied by a positive constant $c$ and the outgoing weights are divided by $c$, the neural network with the new weights will generate exactly the same output as the old one for an arbitrary input. Conventional SGD optimizes ReLU neural networks in weight space. However, it is clear that weight vector is not positively scale-invariant. This mismatch may lead to problems during the optimization process (Neyshabur et al., 2015).

Then, a natural question is: *can we construct a new vector space that is positively scale-invariant and sufficient to represent ReLU neural networks so as to better facilitate the optimization process* ? In this paper, we provide positive answer to this question.

---

[*]The notation $*$ denotes equal contribution.

[†]This work was done when the author was visiting Microsoft Research Asia.

We investigate the positively scale-invariant space to sufficiently represent ReLU neural networks by the following four steps. Firstly, we define the *positive scaling operators* and show that they form a transformation group (denoted as $\mathcal{G}$). The transformation group $\mathcal{G}$ will induce an equivalence relationship called *positive scaling equivalence*. Then, We found that the values of the paths are invariant to positive scaling operators. Furthermore, we prove that two weight vectors are positively scale-equivalent if and only if the values of the paths in one neural network equal to those in the other neural network, given the signs of some weights unchanged. That is to say, the values of all the paths can sufficiently represent a ReLU neural network. After that, we show that the path vectors are linearly dependent.[1] We define the maximal group of paths which are linearly independent as *basis path*, which corresponds to the basis of the *structure matrix* constituted by the path vectors. Thus, the values of the basis paths are also positively scale-invariant and can sufficiently to represent the ReLU neural networks. We denote the vector whose coordinations are composed by values of basis paths as basis path value vector and call the vector space composed by basis path value vector as $\mathcal{G}$-space. In addition, we prove that the dimension of $\mathcal{G}$-space is "$H$" smaller comparing to the weight space, where $H$ is the total number of hidden units in a multi-layer perceptron (MLP) or feature maps in a convolutional networks (CNN).

To sum up, we find $\mathcal{G}$-space constituted by the values of the basis paths, which is positively scale-invariant, can sufficiently represent the ReLU neural networks, and has a smaller dimension than the vector space of weights.

Therefore, we propose to optimize the ReLU neural networks in its positively scale-invariant space, i.e., $\mathcal{G}$-space. We design a novel stochastic gradient descent algorithm in $\mathcal{G}$-space (abbreviated as $\mathcal{G}$-SGD) to optimize the ReLU neural networks utilizing the gradient with respect to the values of the basis paths. First, we design *skeleton method* to construct one group of the basis paths. Then, we develop inverse-chain rule and weight allocation to efficiently compute the gradient of the values of the basis paths by leveraging the back-propagation method. Please note that by using these techniques, there is very little additional computation overhead for $\mathcal{G}$-SGD in comparison with the conventional SGD.

We conduct experiments to show the effectiveness of $\mathcal{G}$-SGD. First, we evaluate $\mathcal{G}$-SGD of training deep convolutional networks on benchmark datasets and demonstrate that $\mathcal{G}$-SGD achieves clearly better performance than baseline optimization algorithms. Second, we empirically test the performance of $\mathcal{G}$-SGD with different degrees of positive scale-invariance. The experimental results show that the higher the positive scale-invariance is, the larger the performance improvement of $\mathcal{G}$-SGD over SGD. This is consistent with that, the positive scale-invariance in weight space will negatively influence the optimization and our proposed $\mathcal{G}$-SGD algorithm can effectively solve this problem.

## 2 BACKGROUNDS

### 2.1 RELATED WORKS

There have been some prior works that study the positively scale-invariant property of ReLU networks and design algorithms that are positively scale-invariant. For example, Badrinarayanan et al. (2015) notice the positive scale-invariance in ReLU netowrks, and inspired by this, they design algorithms to normalize gradients by layer-wise weight norm. Du et al. (2018) study the gradient flow in MLP or CNN models with linear, ReLU or Leaky ReLU activation, and prove the squared norms of gradient across different layers are automatically balanced and remained invariant in gradient descent with infinitesimal step size. In our work, we do not care whether the models are balanced or not. Besides, many other optimization algorithms also have positively scale-invariant property such as Newton's method and natural gradient descent. The most related work is Path-SGD (Neyshabur et al., 2015), which also considers the geometry inspired by path norm. This work is different from ours: 1) they regularize the gradient in weight space by path norm while we optimize the loss function directly in a positively scale-invariant space; 2) they do not consider the dependency between paths and it's hard for them to compute the exactly path-regularized gradients. Different from the previous works, we propose to directly optimize the ReLU networks in its positively scale-invariant space, instead of

---

[1] A path vector is represented by one element in $\{0, 1\}^m$, where $m$ is the number of weights. Please check the details in Section 2.2.

optimizing in the weight space which is not positive scale-invariant. To the best of our knowledge, at the first time, we solve this mismatch by theoretical analysis and an effective and efficient algorithm.

## 2.2 ReLU Neural Networks

Let $N_w(x) : \mathcal{X} \to \mathcal{Y}$ denote a $L$-layer multi-layer perceptron (MLP) with weight $w \in \mathcal{W} \subset \mathbb{R}^m$, the input space $\mathcal{X} \subset \mathbb{R}^d$ and the output space $\mathcal{Y} \subset \mathbb{R}^K$. In the $l$-th layer ($l = 0, \cdots, L$), there are $h_l$ nodes. It is clear that, $h_0 = d, h_L = K$. We denote the $i_l$-th node and its value as $O_{i_l}^l$ and $o_{i_l}^l$, respectively. We use $w^l$ to denote the weight matrix between layer $l-1$ and layer $l$, and use $w^l(i_{l-1}, i_l)$ to denote the weight connecting nodes $O_{i_{l-1}}^{l-1}$ and $O_{i_l}^l$. The values of the nodes are propagated as $o^l = \sigma((w^l)^T o^{l-1})$, where $\sigma(\cdot) = max(\cdot, 0)$ is the ReLU activation function. We use $(i_0, \cdots, i_L)$ to denote the path starting from input feature node $O_{i_0}^0$ to output node $O_{i_L}^L$ passing though hidden nodes $O_{i_1}^1, \cdots, O_{i_{L-1}}^{L-1}$.

We can also regard the network structure as a directed graph $(\mathcal{O}, E)$, where $\mathcal{O} = \{O_1, \cdots, O_{H+d+K}\}$ is the set of nodes where $H$ denotes the number of hidden nodes and $E = \{e_{ij}\}$ denote the set of edges in a network where $e_{ij}$ denotes the edge pointing to $O_j$ from nodes $O_i$. We use $w_e, e \in E$ to denote the weight on edge $e$. If $|E| = m$, the weights compose a vector $w = (w_1, \cdots, w_m)^T$. We define a path as a vector $p = (p_1, \cdots, p_m)^T$ and if the edge $e$ is contained in path $p$, $p_e = 1$; otherwise $p_e = 0$. Because a path crosses $L$ edges for an $L$-layer MLP, there are $L$ elements with value 1 and others elements with value 0. Using these notations, the value of path $p$ can be calculated as $v_p(w) = \prod_{i=1}^m w_i^{p_i}$ and the activation status of path $p$ can be calculated as $a_p(x; w) = \prod_{j:p_{e_{ij}}=1} \mathbb{I}(o_j(x; w) > 0)$. We denote the set composed by all paths as $\mathcal{P}$ and the set composed by paths which contain edge connecting the $i_0$-th input node and the $k$-th output node as $\mathcal{P}^{i_0,k}$. Thus, the output can be computed as follows:

$$N_w^k(x) = \sum_{i_0=1}^d \sum_{p \in \mathcal{P}^{i_0,k}} v_p(w) \cdot a_p(x; w) \cdot x_{i_0}. \tag{1}$$

## 3 Positively Scale-invariant Space of ReLU Networks

In this section, we first define positive scaling transformation group and the equivalence class induced by this group. Then we study the invariant space under positive scaling transformation group of ReLU networks and study its dimension.

### 3.1 Positive Scaling Transformation Group

We formally define the positive scaling operator. We first define a node positive scaling operator $g_{c,O}(w) : \mathcal{W} \to \mathcal{W}$ with constant $c > 0$ and one hidden node $O$ as

$$\tilde{w} = g_{c,O_{i_l}^l}(w),$$

where $\tilde{w}^l(i_{l-1}, i_l) = c \cdot w^l(i_{l-1}, i_l)$ for $i_{l-1} = 1, \cdots, h_{l-1}$; $\tilde{w}^{l+1}(i_l, i_{l+1}) = \frac{1}{c} \cdot w^{l+1}(i_l, i_{l+1})$ for $i_{l+1} = 1, \cdots, h_{l+1}$; and values of other elements of $\tilde{w}$ are the same with $w$.

**Definition 3.1** (*positive scaling operator*) *Suppose that $\{O_1, \cdots, O_H\}$ is the set of all the hidden nodes in the network where $H$ denotes the number of hidden nodes. A positive scaling operator $g_{(c_1, \cdots, c_H)}(\cdot) : \mathcal{W} \to \mathcal{W}$ with $c_1, \cdots, c_H \in \mathbb{R}^+$ is defined as*

$$g_{(c_1, \cdots, c_H)}(\cdot) := g_{c_1, O_1} \circ g_{c_2, O_2} \circ \cdots \circ g_{c_H, O_H}(\cdot),$$

*where $\circ$ denotes function composition.*

We then collect all the $g_{(c_1, \cdots, c_H)}(\cdot)$ together to form a set $\mathcal{G} := \{g_{(c_1, \cdots, c_H)}(\cdot) : c_1, \cdots, c_H \in \mathbb{R}^+\}$. It is easy to check that $\mathcal{G}$ together with the operation "$\circ$" is a group which is called *positive scaling transformation group*, and we call the group action of $\mathcal{G}$ on $\mathcal{W}$ as $\mathcal{G}$-action. (Please refer to Section 8 in Appendix.) Clearly, if there exists an operator $g \in \mathcal{G}$ to make $w = g(w')$, ReLU networks $N_w$ and $N_{w'}$ will generate the same output for any fixed input $x$. We define the positive scaling equivalence induced by $\mathcal{G}$-action.

**Definition 3.2** *Consider two ReLU networks with weights $w, w' \in \mathcal{W}$ and the positive scaling transformation group $\mathcal{G}$. We say $w$ and $w'$ are positively scale-equivalent if $\exists g \in \mathcal{G}$ such that $w = g(w')$, denote as $w \sim_{\mathcal{G}} w'$.*

Given $\mathcal{G}$-action on $\mathcal{W}$, the equivalence relation "$\sim_{\mathcal{G}}$" partitions $\mathcal{W}$ into $\mathcal{G}$-equivalent classes. The following theorem shows that the sufficient and necessary condition for ReLU networks in the same equivalent class is that they have the same values and activation status of paths.

**Theorem 3.3** *Consider two ReLU neural networks with weights $w, w' \in \mathcal{W}$. We have that $w \sim_{\mathcal{G}} w'$ iff for $\forall$ path $p \in \mathcal{P}$ and any fixed input $x \in \mathcal{X}$, we have $v_p(w) = v_p(w')$ and $a_p(x; w) = a_p(x; w)$.*

Invariant variables for a group action are important and widely studied in group theory and geometry. We say a function $f : \mathcal{W} \to \mathbb{R}$ is invariant variable of $\mathcal{G}$-action if $f(w) = f(g(w)), \forall g \in \mathcal{G}$. Based on Theorem 3.3, a direct corollary is that values and activation status of paths are invariant variables under $\mathcal{G}$-action. Considering that 1) values of paths are $\mathcal{G}$-invariant variables while the weights aren't; 2) values of paths together with the activation status determines an positive scale-equivalent class, and are sufficient to determine the loss, we propose to optimize the values of paths instead of weights.

## 3.2 Positively Scale-Invariant Space and Its Dimension

Although Theorem 3.3 shows the values of paths are invariant variables under $\mathcal{G}$-action, we find that the paths have inner-dependency and therefore their values and activation statuses are not independent. Let us consider the example in Figure 1. The values of paths have the relationship $v_{p^4}(w) = \frac{v_{p^2}(w) \cdot v_{p^3}(w)}{v_{p^1}(w)}$. Using the vector representation of paths that is described in section 2.2, we find that their path vectors follow the relationship $p^4 = p^2 + p^3 - p^1$, which means that they are not linearly independent.

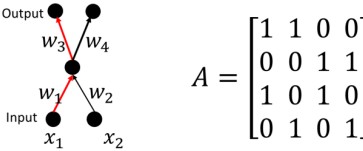

$$A = \begin{bmatrix} 1 & 1 & 0 & 0 \\ 0 & 0 & 1 & 1 \\ 1 & 0 & 1 & 0 \\ 0 & 1 & 0 & 1 \end{bmatrix}$$

Figure 1: This is a simple ReLU network with one hidden node. Suppose path values are $v_{p^1}(w) = w_1 w_3, v_{p^2}(w) = w_1 w_4, v_{p^3}(w) = w_2 w_3, v_{p^4}(w) = w_2 w_4$, we can see the inner-dependency between them, i.e., $v_{p^4}(w) = \frac{v_{p^2}(w) \cdot v_{p^3}(w)}{v_{p^1}(w)}$. $A$ is the structure matrix of this example.

For a feedforward ReLU neural network, we suppose that $\mathcal{P} \subset \{0, 1\}^m$ is the set composed by all path vectors. We denote the matrix composed by all paths as $A$ and call it *structure matrix* of ReLU networks. The size of $A$ is $m \times n$ where $n$ is the number of paths. We observe that the paths in matrix $A$ are not linearly independent. Then we study the rank of matrix $A$ and find a maximal linearly independent group of paths.

**Theorem 3.4** *If $A$ is the structure matrix for a ReLU network, then we have $rank(A) = m - H$, where $m$ is the dimension of weight vector $w$ and $H$ is the total number of hidden nodes for MLP (or feature maps for CNN models) with ReLU respectively.*

**Definition 3.5** *(basis path) A set of paths $\mathcal{P}_0 = \{p^1, \cdots, p^{m-H}\}$ which is a subset of $\mathcal{P}$ is called a set of basis paths if $p^1, \cdots, p^{m-H}$ compose a maximal linearly independent group of column vectors in structure matrix $A$.*

We design an algorithm called *skeleton method* to identify basis paths efficiently, which will be introduced in Section 4. For given values of basis paths and structure matrix, the values of $w$ can not be determined unless the values of free variables are fixed (Lay (1997)). Assume $w_{s_1}, \cdots, w_{s_H}$ are selected to be the free variables which are called *free skeleton weights*, we prove that the activation status can be uniquely determined by the values of basis paths if signs of free skeleton weights are fixed. Thus, we have the following theorem which is a modification of Theorem 3.3.

**Theorem 3.6** *Consider two ReLU neural networks with weights $w, w' \in \mathcal{W}$ with the same signs of skeleton weights. We have that $w \sim_{\mathcal{G}} w'$ iff for $\forall p \in \mathcal{P}_0$, we have $v_p(w) = v_p(w')$.*

The detailed proof of Theorem 3.4 and Theorem 3.6 are both depends on Lemma 9.1 in Appendix. In the following context, we always suppose that $\forall w \in \mathcal{W}$ have the same signs of free skeleton weights.

According to Theorem 3.6 and the linear dependency between values of paths, the loss function can be calculated using values of basis paths if signs of free skeleton weights are fixed. We denote the the loss at training instance $(x, y)$ as $l(v; x, y)$ and propose to optimize the values of basis paths. Considering that values of basis paths are obtained through structure matrix $A$, the dimension of the space composed by values of basis paths should be equal to $rank(A)$. Then we define the following space.

**Definition 3.7** *($\mathcal{G}$-space) The $\mathcal{G}$-space is defined as $V := \{v = (v_{p^1}, \cdots, v_{p^{m-H}}) : v \in (\mathbb{R}/\{0\})^{m-H}\}$.*

We call the space composed by the values of basis paths $\mathcal{G}$-space, which is invariant under transformation group $\mathcal{G}$, i.e., it is composed by invariant variables under $\mathcal{G}$-action. Immediately, we can get the following corollary according to Theorem 3.6.

**Corollary 3.8** *The dimension of $\mathcal{G}$-space is $m - H$, where $m$ is the number of weights and $H$ is the total number of hidden nodes for MLP or the total number of feature maps for CNN.*

We measure the reduction of the dimension for positively scale-invariant space using the invariant ratio $H/m$, thus we can empirically test how severe this equivalence will influence the optimization in weight space.

# 4 ALGORITHM: $\mathcal{G}$-SGD

In this section, we will introduce the $\mathcal{G}$-SGD that optimizes ReLU neural network models in the $\mathcal{G}$-space. This novel algorithm makes use of three methods, named Skeleton Method, Inverse-Chain-Rule (ICR) and Weight-Allocation (WA), respectively, to calculate the gradients w.r.t. basis path vector and project the updates back to weights efficiently (with little extra computation in comparison with standard SGD).

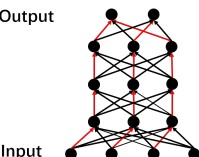

Figure 2: The weights with red color are skeleton weights.

## 4.1 SKELETON METHOD

Before the calculation of gradients in $\mathcal{G}$-space, we first design an algorithm called *skeleton method* to construct skeleton weights and basis paths for MLP whose depth is $L$ and width is $h$.

*1. Construct skeleton weights:* for weight matrix $w^2, \cdots, w^{L-1}$, we select diagonal elements to be the *skeleton weights*. For weight matrix $w^1$, we select the element $w^1(i_1 \mod d, i_1)$ for column $i_1$ with $i_1 = 1, \cdots, h_1$ to be the *skeleton weights*. For weight matrix $w^L$, we select the element $w^L(i_{L-1}, i_{L-1} \mod K)$ for row $i_{L-1}$ with $i_{L-1} = 1, \cdots, h_{L-1}$ to be the *skeleton weights*. We call the rest weights non-skeleton weights. Figure 2 gives an illustration for skeleton weights in a MLP network.

*2. Construct basis paths:* A path which contains at most one non-skeleton weights is a basis path. The proof of this statement could be found in Appendix. For example, in Figure 2, the paths in red color and the paths with only one black weight are basis paths. Beyond that, the paths are non-basis paths.

Once we have basis paths, we can calculate the gradients w.r.t. basis path vector $v_{p^i}$, and iteratively update the model by

$$v_{p^j}^{t+1} = v_{p^j}^t - \eta_t \frac{\partial l(v; S^t)}{\partial v_{p^j}}\bigg|_{v=v^t}, j = 1, \cdots, m - H, \tag{2}$$

where $S^t$ is the mini-batch training data in iteration $t$. For the calculation of the gradients w.r.t. basis path vector, we introduce inverse-chain-rule method in next section.

## 4.2 INVERSE-CHAIN-RULE (ICR) METHOD

The basic idea of the *Inverse-Chain-Rule* method is to connect the gradients w.r.t. weight vector and those w.r.t. basis path vector by exploring the chain rules in both directions. That is, we have,

$$
(\frac{\partial l(w;x,y)}{\partial w_1}, \cdots, \frac{\partial l(w;x,y)}{\partial w_m}) = (\frac{\partial l(v;x,y)}{\partial v_{p^1}}, \cdots, \frac{\partial l(v;x,y)}{\partial v_{p^{m-H}}}) \cdot \begin{bmatrix} \frac{\partial v_{p^1}}{\partial w_1} & \cdots & \frac{\partial v_{p^1}}{\partial w_m} \\ & \cdots & \\ \frac{\partial v_{p^{m-H}}}{\partial w_1} & \cdots & \frac{\partial v_{p^{m-H}}}{\partial w_m} \end{bmatrix} \quad (3)
$$

We first compute the gradients w.r.t. weights, i.e., $\frac{\partial l(w;x,y)}{\partial w_i}$ for $i = 1, \cdots, m$ using standard back propagation. Then we solve Eqn.(3) to obtain the gradients w.r.t. basis paths, i.e., $\frac{\partial l(v;x,y)}{\partial v_j}$ for $j = 1, \cdots, m - H$. We denote matrix at the right side of Eqn.(3) as $G$. Given the following facts: (1) $\frac{\partial v_p}{\partial w_e} = \frac{v_p}{w_e}$ if the edge $e$ is contained in path $p$, otherwise 0; (2) according to the skeleton method, each non-skeleton weight will be contained in only one basis path, which means there is only one non-zero element in each column corresponding to non-skeleton weights in $G$, $G$ is sparse and thus the solution of Eqn.(3) is easy to obtain.

## 4.3 WEIGHT-ALLOCATION (WA) METHOD

After the values of basis paths are updated by SGD, in a new iteration, we employ ICR again by leveraging BP with the new weight. Thus, we need to project the updates on basis paths back to the updates of weights.

We define the *path-ratio* of $p^j$ at iteration $t$ as $R^t(p^j) := v_{p^j}^t / v_{p^j}^{t-1}$ and the *weight-ratio* of $w_i$ at iteration $t$ as $r^t(w_i) := w_i^t / w_i^{t-1}$. Assume that we have already obtained the path-ratio for all the basis paths $R^{t+1}(p^j)$ according to ICR method and the SGD update rule. Then we want to project the path-ratios onto the weight-ratios. We use the notation $\odot$ to denote the operation $w \odot p^j = \prod_{i=1}^m w_i^{p_i^j}$ Because we have $v_{p^j}(w) = w \odot p^j$, the weight-ratios obtained after the projection should satisfy the following relationship: $R^t(p^j) = r^t(w) \odot p^j$. Generalize the operator $\odot$ from vectors to matrices [2], we have the following relationship:

$$
(R^t(p^1), \cdots, R^t(p^{m-H})) = (r^t(w_1), \cdots, r^t(w_m)) \odot A', \quad (4)
$$

where the matrix $A' = (p^1, \cdots, p^{m-H})$. According to this relationship, we design *Weight-Allocation Method* to project the path-ratio to weight-ratio as described below. Suppose that $w_1, \cdots, w_H$ are the free skeleton weights. We first add $H$ elements with value 1 at the beginning in vector $(R^t(p^1), \cdots, R^t(p^{m-H}))$ to get a new $m$-dimensional vector. Then we append $H$ columns in matrix $A'$ to get a new matrix $\tilde{A}$ as $\tilde{A} = [B, A']$ with $B = [I, \mathbf{0}]^{\mathbf{T}}$ where $I$ is an $H \times H$ identity matrix with diagonal elements 1 and $\mathbf{0}$ is an $H \times m$ zero matrix with all elements 0. Then it is easy to prove that $rank(\tilde{A}) = m$ and we have the following relationship:

$$
(1, \cdots, 1, R^t(p^1), \cdots, R^t(p^{m-H})) = (r^t(w_1), \cdots, r^t(w_m)) \odot \tilde{A}. \quad (5)
$$

We can get the weight-ratio by $\odot \ \tilde{A}^{-1}$ on both sides of Eq.(5). Because we have $(r^t(w_1), \cdots, r^t(w_m)) \odot \tilde{A} \odot \tilde{A}^{-1} = (r^t(w_1), \cdots, r^t(w_m))$ (refer to Section 8 in Appendix), we have

$$
(r^t(w_1), \cdots, r^t(w_m)) = (1, \cdots, 1, R^t(p^1), \cdots, R^t(p^{m-H})) \odot \tilde{A}^{-1}. \quad (6)
$$

After the projection, we can see that weight-ratios of free skeleton weights equal 1 which means that free skeleton weights will not be changed during the training process. According to the skeleton method again, $\tilde{A}$ is a sparse matrix and it is easy to calculate its inverse.

Actually, the projection method is not unique. Although we choose one special projection which fixed values of free skeleton weights in the Weight-Allocation method, we prove that different projection methods will results in the same updates in $\mathcal{G}$-space if they don't change the signs of free skeleton weights.

**Theorem 4.1** *Suppose that there are two different projections $\mathcal{T}_1$ and $\mathcal{T}_2$ that project the path-ratio to weight-ratio. If the projection will not change the signs of free skeleton weights, the values of basis paths will keep the same at every iteration for two $\mathcal{G}$-SGD processes that are initialized with the same values of basis paths and use different projections $\mathcal{T}_1$ and $\mathcal{T}_2$ in WA method respectively.*

---

**Algorithm 1** $\mathcal{G}$-SGD

---

**Require:** initialization $w^0$, learning rate $\eta_t$, training data set $D$.

  1. Construct skeleton weights using skeleton method and construct $\tilde{A}^{-1}$ according to skeleton method and WA method.

  **for** $t = 1, \cdots, T$ **do**

    2. Implement feed forward process and back propagation process to get $\frac{\partial l(w;S^t)}{\partial w_i}\big|_{w=w_t}$.

    3. Calculate $\frac{\partial l(v;x,y)}{\partial v_{p^j}}$ for $j = 1, \cdots, m - H$ according to Eqn.(3).

    4. Using SGD to update the values of basis paths: $v_{p^j}^{t+1} = v_{p^j}^t - \eta_t \cdot \frac{\partial l(v;x,y)}{\partial v_{p^j}}$.

    5. Calculate path-ratios: $R^t(p^j) = v_{p^j}^{t+1}/v_{p^j}^t$.

    6. Calculate weight-ratio $r^t(w_i)$ using $\tilde{A}^{-1}$ according to Eqn.(6).

    7. Update the weights as $w_i^{t+1} = w_i^t \cdot r^{t+1}(w_i)$.

  **end for**

**Ensure:** $w^T$.

---

Please note by combining the ICR and WA methods, we can obtain the explicit update rule for $\mathcal{G}$-SGD, which is concluded in Algorithm 1. In this way, we obtain the correct gradients. The extra computational complexity of the ICR and WA methods are far lower than that of forward and backward propagation, and can therefore be neglected in practice.

## 5   EXPERIMENTS

In this section, we first evaluate the performance of $\mathcal{G}$-SGD on training deep convolutional networks and verify that if our proposed algorithm outperforms other baseline methods. Then we investigate the influence of positive scaling invariance on the optimization in weight space, and examine whether optimization in $\mathcal{G}$ space brings performance gain. At last, we compare $\mathcal{G}$-SGD with Path-SGD (Neyshabur et al., 2015) and show the necessity of considering the dependency between paths. All experiments are averaged over 5 independent trials if without explicit note.

### 5.1   DEEP CONVOLUTIONAL NETWORK

In this section, we apply our $\mathcal{G}$-SGD to image classification tasks and conduct experiments on CIFAR-10 and CIFAR-100 (Krizhevsky & Hinton, 2009). In our experiments, we employ the original ResNet architecture described in (He et al., 2016a). Specifically, there is no positive scaling invariance across residual blocks since the residual connections break down the structure matrix described in Section 3.2, we target the invariance in each residual block. For better comparison, we also conduct our studies on a stacked deep CNN described in He et al. (2016a) (refer to PlainNet), and target the positive scaling invariance across all layers. We train 34 layers ResNet and PlainNet models on the datasets following the training strategies in the original paper, and compare the performance between $\mathcal{G}$-SGD[3] and vanilla SGD algorithm. The detailed training strategies could be found in Appendix. In this section, we focus on the performance of different optimization algorithms, and will discuss the combination of $\mathcal{G}$-SGD and regularization in Appendix.

Table 1: Classification error rate (%) on image classification task.

|  |  | C10 | C100 |
|---|---|---|---|
| Plain-34 | SGD | 7.76 ($\pm$0.17) | 36.41($\pm$0.54) |
|  | $\mathcal{G}$-SGD | **7.00** ($\pm$0.10) | **30.74** ($\pm$0.29) |
| ResNet-34 | SGD | 7.13 ($\pm$0.22) | 28.60($\pm$0.51) |
|  | $\mathcal{G}$-SGD | **6.66** ($\pm$0.13) | **27.74** ($\pm$0.24) |

As shown in Figure 3 and Table 1, our $\mathcal{G}$-SGD clearly outperforms SGD on each network and each dataset. To be specific, 1) both the lowest training loss and best test accuracy are achieved by

---

[2]For strict description of operation "$\odot$", please refer to Section 8 in Appendix.

[3]Batch normalization is widely used in modern CNN models. Please refer to Appendix for the combination of $\mathcal{G}$-SGD and batch normalization.

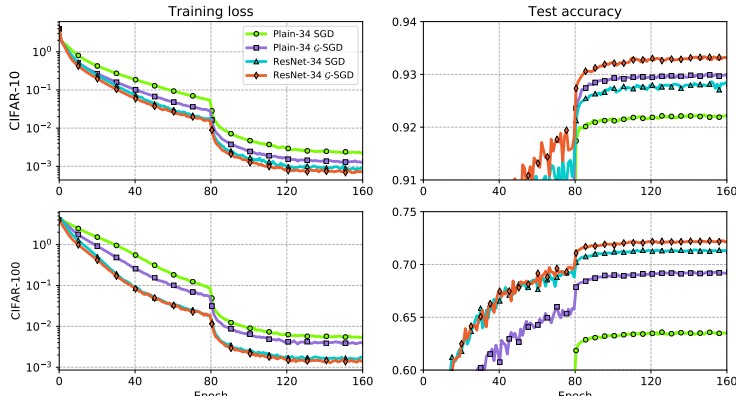

Figure 3: Training loss and test accuracy w.r.t. the number of effective passes on PlainNet and ResNet.

ResNet-34 with $\mathcal{G}$-SGD on both datasets, which indicates that $\mathcal{G}$-SGD indeed helps the optimization of ResNet model; 2) Since $\mathcal{G}$-SGD can eliminate the influence of positive scaling invariance across all layers of PlainNet, we observe the performance gain on PlainNet is larger than that on ResNet. For PlainNet model, $\mathcal{G}$-SGD surprisingly improves the accuracy numbers by 0.8 and 5.7 for CIFAR-10 and CIFAR-100, respectively, which verifies both the improper influence of positive scaling invariance for optimization in weight space and the benefit of optimization in $\mathcal{G}$ space. Moreover, Plain-34 trained by $\mathcal{G}$-SGD achieves even better accuracy than ResNet-34 trained by SGD on CIFAR-10, which shows the influence of invariance on optimization in weight space as well.

## 5.2 THE INFLUENCE OF INVARIANCE

In this section, we study the influence of invariance on the optimization for ReLU Networks. As proved in Section 3, the dimension of weight space is larger than $\mathcal{G}$-space by $H$, where $H$ is the total number of the hidden nodes in a MLP or the feature maps in a CNN. We define the invariant ratio as $H/m$. We train several 2-hidden-layer MLP models on Fasion-MNIST (Xiao et al., 2017) with different number of hidden nodes in each layer, and analyze the performance gap $\Delta$ between the models optimized by $\mathcal{G}$-SGD and SGD. The detailed training strategies and network structures could be found in Appendix.

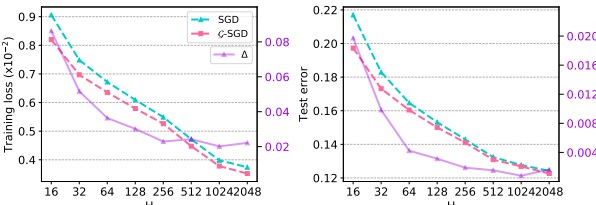

Figure 4: Training loss and test error on MLPs. The invariant ratio decreases as $H$ increases.

From Figure 4, we can see that, 1) for each number of $H$, $\mathcal{G}$-SGD clearly outperforms SGD on both training loss and test error, which verifies our claim that optimization loss function in $\mathcal{G}$ space is a better choice; 2) as $H$ increases, the invariant ratio decreases (because $m$ also increases for MLP) and $\Delta$ gradually decreases as well, which provides the evidence for that the positive scaling invariance in weight space indeed improperly influences the optimization.

## 5.3 COMPARISON WITH PATH-SGD

In this section, we compare the performance of Path-SGD and that of $\mathcal{G}$-SGD. As described in Section 2.1, Path-SGD also consider the positive scaling invariance, but 1) instead of optimizing the loss function in $\mathcal{G}$-space, Path-SGD regularizes optimization by path norm; 2) Path-SGD ignores the dependency among the paths. We extend the experiments in Neyshabur et al. (2015) to $\mathcal{G}$-SGD

without unbalance initialization, and conduct our studies on MNIST and CIFAR-10 datasets. The detailed training strategies and description of network structure can be found in Appendix.

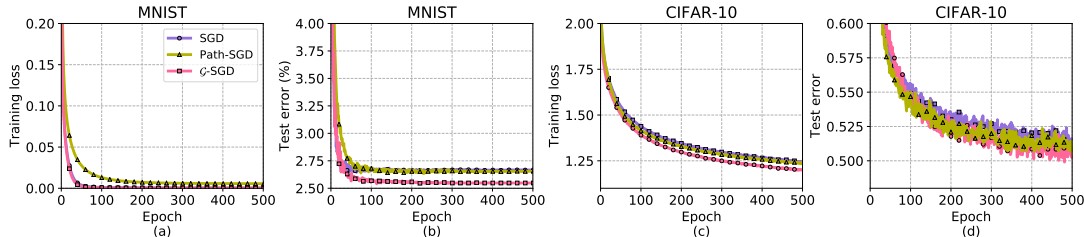

Figure 5: Performance of MLP models with Path-SGD and $\mathcal{G}$-SGD.

As shown in Figure 5, while Path-SGD achieves better or equally good test accuracy and training loss than SGD for both MNIST and CIFAR10 datasets, $\mathcal{G}$-SGD achieves even better performance than Path-SGD, which is consistent with our theoretical analysis that considering the dependency between the paths and optimizing in $\mathcal{G}$-space bring benefit.

## 6 CONCLUSION

In this paper, we study the $\mathcal{G}$-space for ReLU neural networks and propose a novel optimization algorithm called $\mathcal{G}$-SGD. We study the positive scaling operators which forms a transformation group $\mathcal{G}$ and prove that the value vector of all the paths is sufficient to represent the neural networks. Then we show that one can identify basis paths and prove that the linear span of their value vectors (denoted as $\mathcal{G}$-space) is an invariant space with lower dimension under the positive scaling group. We design $\mathcal{G}$-SGD algorithm in $\mathcal{G}$-space by leveraging back-propagation. We conduct extensive experiments to verify the empirical effectiveness of our proposed approach. In the future, we will examine the performance of $\mathcal{G}$-SGD on more large-scale tasks.

ACKNOWLEDGMENTS

This work is partially supported by National Center for Mathematics and Interdisciplinary Sciences (NCMIS).

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

APPENDIX: $\mathcal{G}$-SGD: OPTIMIZING RELU NEURAL NETWORKS IN ITS POSITIVELY SCALE-INVARIANT SPACE

The Appendix document is composed of examples of skeleton weights and basis paths for different MLP structures, proofs of propositions, lemmas and theorems and the additional information about the experiments in the paper Optimization of ReLU Neural Networks using $\mathcal{G}$-Stochastic Gradient Descent .

## 7 NOTATIONS

Table 2: Notations

| Notations | Object |
|---|---|
| $m$ | dimension of weight space |
| $H$ | total number of hidden nodes or feature maps |
| $n$ | total number of paths |
| $m - H$ | total number of basis paths and dimension of $\mathcal{G}$-space |
| $\mathcal{W} \subset \mathbb{R}^m$ | weight vector space |
| $w = (w_1, \cdots, w_m)$ | weight vector with $m = \sum_{l=1}^{L} h_{l-1} h_l$ for MLP |
| $w^l$ | weight matrix at layer $l$ with size $h_{l-1} \times h_l$ for MLP |
| $w^l(i_{l-1}, i_l)$ | weight element in matrix $w^l$ at position $(i_{l-1}, i_l)$ |
| $O_{i_l}^l$ | the $i_l$-th hidden node at layer $l$ |
| $E = \{e_{ij}\}$ | the set of edges in neural network model |

Table 3: Index

| Index | Range | Object |
|---|---|---|
| $l$ | $\{1, \cdots, L\}$ | index of layer |
| $i_l$ | $\{1, \cdots, h_l\}$ | index of hidden nodes at $l$-layer |
| $(i_L, i_{L-1}, \cdots, i_0)$ | $i_l \in [h_l], l \in [L]$ | explicit index of path |
| $p$ | $\mathcal{P}$ | path |
| $p^i$ | $\{p^1, \cdots, p^{m-H}\} = \mathcal{P}_0$ | basis path |
| $s_j$ | $\{s_1, \cdots, s_H\} \subset \{1, \cdots, m\}$ | free skeleton weight |

Table 4: Mathematical Notations

| Notation | Meaning |
|---|---|
| # | the number of |
| / | division |
| $\circ$ | function composition |

## 8 SOME CONCEPTS IN ABSTRACT ALGEBRA

**Definition 8.1** (*Transformation group*) *Suppose that $\mathcal{G}$ is a set of transformations, and $\circ$ is an operation defined between the elements of $\mathcal{G}$. If $\mathcal{G}$ satisfies the following conditions: 1) (operational closure) for any two elements $g_1, g_2 \in \mathcal{G}$, it has $g_1 \circ g_2 \in \mathcal{G}$; 2) (associativity) for any three elements $g_1, g_2, g_3 \in \mathcal{G}$, it has $(g_1 \circ g_2) \circ g_3 = g_1 \circ (g_2 \circ g_3)$; 3) (unit element) there exists unit element $e \in \mathcal{G}$, so that for any element $g \in \mathcal{G}$, there is $g \circ e = g$; 4) (inverse element) for any element $g \in \mathcal{G}$, there exists an inverse element $g^{-1} \in \mathcal{G}$ of $g$ such that $g \circ g^{-1} = e$. Then, $\mathcal{G}$ together with the operation "$\circ$" is called a transformation group.*

**Definition 8.2** (*Group action*) *If $\mathcal{G}$ is a group and $\mathcal{W}$ is a set, then a (left) group action $\phi_{\mathcal{G}, \mathcal{W}}$ of $\mathcal{G}$ on $\mathcal{W}$ is a function $\phi_{\mathcal{G}, \mathcal{W}} : \mathcal{G} \times \mathcal{W} \to \mathcal{W}$ that satisfies the following two axioms (where we denote $\phi(g, w)$ as $g \cdot w$): 1) (identity) $e \cdot w = w$; 2) (compatibility) $(g \circ h) \cdot w = g \circ (h \cdot w)$ for all $g, h \in \mathcal{G}$ and all $w \in \mathcal{W}$.*

**Definition 8.3 (*Operation* $\odot$)** *We define $\odot$ as a right group action for matrix $W_{m \times s} = [w_{ij}]_{i=1,\cdots,m;j=1\cdots,s}$ with $w_{ij} \in (\mathbb{R}/\{0\})$ as:*

$$W \odot A = \left[ \begin{array}{ccc} \prod_{j=1}^s sgn(w_{1j}) \cdot |w_{1j}|^{a_{j1}} & \cdots & \prod_{j=1}^s sgn(w_{1j}) \cdot |w_{1j}|^{a_{jn}} \\ \cdots & \cdots & \cdots \\ \prod_{j=1}^s sgn(w_{mj}) \cdot |w_{mj}|^{a_{j1}} & \cdots & \prod_{j=1}^s sgn(w_{mj}) \cdot |w_{mj}|^{a_{jn}} \end{array} \right],$$

*where $A_{s \times n} = [a_{jk}]_{i=1,\cdots,s;k=1,\cdots,n}$ is a matrix with $a_{ij} \in \mathbb{R}$.*

According to the definition, we can prove that $W \odot A \odot A^{-1} = W$ if $A$ is a square matrix. We use $A^{-1} = [\tilde{a}_{kz}]_{k=1,\cdots,s;z=1,\cdots,s}$. Then the element at the $b$-th row and $c$-th column of $W \odot A \odot A^{-1}$ is calculated as $\prod_{j=1}^s sgn(w_{bj}) \cdot \prod_{k=1}^s (\prod_{j=1}^s \cdot|w_{bj}|^{a_{jk}})^{\tilde{a}_{kc}} = \prod_{j=1}^s sgn(w_{bj}) \cdot \prod_{j=1}^s \cdot|w_{bj}|^{\sum_{k=1}^s a_{jk}\tilde{a}_{kc}}$. When $j = c$, $\sum_{k=1}^s a_{jk}\tilde{a}_{kc} = 1$; otherwise, $\sum_{k=1}^s a_{jk}\tilde{a}_{kc} = 0$. Thus we have $\prod_{j=1}^s sgn(w_{bj}) \cdot \prod_{k=1}^s (\prod_{j=1}^s \cdot|w_{bj}|^{a_{jk}})^{\tilde{a}_{kc}} = w_{bc})$. Thus we have prove the claim: $W \odot A \odot A^{-1} = W$. Thus Eq.(6) in the main paper is established.

# 9 PROOFS OF THEORETICAL RESULTS

In this section, we will provide proofs of the lemma and theorems in Section 3 of the main paper.

## 9.1 PROOF OF THEOREM 3.3

**Theorem 3.3:** *Consider two ReLU neural networks with weights $w, w' \in \mathcal{W}$. We have that $w \sim w'$ iff for $\forall$ path $p$ and $\forall$ input $x \in \mathcal{X}$, we have $v_p(w) = v_p(w')$ and $a_p(w, x) = a_p(w', x)$.*

*Proof:* For the necessity, if $w \sim w'$, then there exist a positive scaling operator $g(\cdot)$ to make $g(w') = w$. We use $i_l$ to denote the node index of nodes in layer-$l$ and $i_l \in [h_l]$. Then we have $w_l(i_{l-1}, i_l) = \frac{1}{c_{i_{l-1}}^{l-1}} \cdot c_{i_l}^l \cdot w_l'(i_{l-1}, i_l)$ for $l = 2, \cdots, L-1$, because each weight may be modified by the operators of its connected two nodes $g_{c_{i_{l-1}}^{l-1}, O_{i_{l-1}}^{l-1}}$ and $g_{c_{i_l}^l, O_{i_l}^l}$. Thus $v_p(w) = v_p(w')$ is satisfied because

$$v_p(w') = \prod_{l=1}^L w_l'(i_{l-1}, i_l) = c_1^1 w_1'(i_0, i_1) \cdot \left( \prod_{l=2}^{L-1} \frac{1}{c_{i_{l-1}}^{l-1}} \cdot c_{i_l}^l w_l'(i_{l-1}, i_l) \cdot \right) \cdot \frac{1}{c_{i_{L-1}}^{L-1}} w_L'(i_{L-1}, i_L) \quad (7)$$

$$= \prod_{l=1}^L w_l(i_{l-1}, i_l) = v_p(w). \quad (8)$$

Next we need to prove that $a_p(w, x) = a_p(w', x)$ is also satisfied. Because the value of the activation is determined by the sign of $o^l, l = \{1, \cdots, L-1\}$, we just need to prove that

$$[o_{w,1}^l(x), \cdots, o_{w,h_l}^l(x)] = [c_1^l \cdot o_{w',1}^l(x), \cdots, c_{h_l}^l \cdot o_{w',h_l}^l(x)],$$

where $c_{i_l}^l, i_l \in [h_l], l \in [L-1]$ are positive numbers. We prove it by induction.

(1) For $o^1$ of a $L$-layer MLP ($L > 2$): Suppose that $\sigma(\cdot)$ is a ReLU activation function. For the $i_1$-th hidden node, we have

$$o_{w,i_1}^1 = \sigma \left( \sum_{i_0=1}^{h_0} w_1(i_0, i_1)x_{i_0} \right) = \sigma \left( \sum_{i_0=1}^{h_0} c_{i_1}^1 \cdot w_1'(i_0, i_1)x_{i_0} \right) = c_{i_1}^1 \cdot \sigma \left( \sum_{i_0=1}^{h_0} w_1'(i_0, i_1)x_{i_0} \right) = c_{i_1}^1 \cdot o_{w',i_1}^1. \quad (9)$$

(2) For $o^l$ of the $L$-layer MLP ($l > 2$): Suppose that

$$[o_{w,1}^j(x), \cdots, o_{w,h_j}^j(x)] = [c_1^j \cdot o_{w',1}^j(x), \cdots, c_{h_j}^j \cdot o_{w',h_j}^j(x)], j = \{1, \cdots, l-1\}.$$

Then we have

$$o_{w,i_l}^l = \sigma \left( \sum_{i_{l-1}=1}^{h_{l-1}} w_l(i_{l-1}, i_l)o_{w,i_{l-1}}^{l-1}(x) \right) = \sigma \left( \sum_{i_{l-1}=1}^{h_{l-1}} \frac{1}{c_{i_{l-1}}^{l-1}} \cdot c_{i_l}^l \cdot w_l'(i_{l-1}, i_l) \cdot c_{i_{l-1}}^{l-1} \cdot o_{w',i_{l-1}}^{l-1}(x) \right)$$

$$= c_{i_l}^l \cdot o_{w',i_l}^l.$$

Thus we finish the proof of necessity.

For the sufficiency, we need to prove that if $v_p(w) = v_p(w')$ and $a_p(w, x) = a_p(w', x)$ for $\forall$ path $p$ and $\forall$ input $x \in \mathcal{X}$, there exists $g \in \mathcal{G}$ to make $w' = g(w)$. First, for hidden node $O_{i_1}^1$ in layer-1, we claim that their incoming weights satisfy the following relationship:

$$w_1'(i_0, i_1) = c_{i_1}^1 \cdot w_1(i_0, i_1), \forall i_0 \in [h_0].$$

Because there exist path $p^{z_1}, \cdots, p^{z_{h_0}} \in \mathcal{P}$ to make

$$v_{p^{z_1}}(w) : v_{p^{z_2}}(w) : \cdots : v_{p^{z_{h_0}}}(w) = w_1(1, i_1) : w_1(2, i_1) : \cdots : w_1(h_0, i_1),$$

and $v_p(w) = v_p(w')$ for any $p$, we have

$$w_1(1, i_1) : w_1(2, i_1) : \cdots : w_1(h_0, i_1) = w_1'(1, i_1) : w_1'(2, i_1) : \cdots : w_1'(h_0, i_1).$$

Then the claim is established. Then we prove each $c_{i_1}^1$ is positive. If there exist a constant $c_{i_1}^1$ is negative, then $o_{i_1}^1(w, x) \neq o_{i_1}^1(w', x)$. If $o_{i_l}^l(w, x) > 0$, we have $o_{i_l}^l(w', x) = 0$. Here we assume that $x \in \mathcal{X}$ where $\mathcal{X}$ is a compact set to make that $a_p(w, x) = a_p(w', x), \forall p, \forall x$ is equivalent to $sgn(o_{i_l}^l(w, x)) = sgn(o_{i_l}^l(w', x)), \forall x, \forall l = 1, \cdots, L$. Thus all the $c_{i_1}^1$ for $i_1 = 1, \cdots, h_1$ are positive. Then we use the operator $g_{c_1^1, O_1^1} \circ \cdots \circ g_{c_{h_1}^1, O_{h_1}^1}(w)$ to make the two networks with the same weights at layer 1. Then based on the networks with the same weights at layer 1, we gradually deal with the weights in other layers from layer 2 to the last layer using similar techniques as layer 1. We can finally construct $g \in \mathcal{G}$. Thus we finish the proof of the sufficiency.

$\square$

## 9.2 Proof of Theorem 3.4 and Theorem 3.6

In order to prove Theorem 3.4 and Theorem 3.6, we need to prove that there exist a group of paths which are independent and can represent other paths, and the activation status can be calculated using values of basis paths and signs of free skeleton weights. In order to simplify the proof, we leverage the basis paths constructed by skeleton method. We only show the proof of the following lemma, from which we can easily get Theorem 3.4 and Theorem 3.6.

**Lemma 9.1** *The paths selected by skeleton method are basis paths and $a_p(w, x)$ can be calculated using signs of free skeleton weights and the values of basis paths in a recursive way.*

*Proof sketch:* Let us consider the matrix $A' = (p^1, \cdots, p^{m-H})$ composed by basis paths constructed by skeleton method:

$$A' = \begin{pmatrix} I & 0 \\ B_1 & B_2 \end{pmatrix} \tag{10}$$

There is an identity matrix with size $z \times z$ where $z$ is the number of skip skeleton paths. This identity matrix means that $w_i$ is a non-skeleton weight and is contained in $p^i$, $i \leq z$. Thus, through the row transformation of the matrix, $B_1$ can be transformed to zero matrix. According to skeleton method, column vectors in $B_2$ are independent because skeleton weight will only appear in one all-basis path. Thus the independent property has been proved. Furthermore, by leveraging the structure of matrix $A'$, it is easily to check that for a non-skeleton path $p$, it can be calculated as $p = \sum_{i=1}^z \alpha_i p^i - \sum_{j=z+1}^{m-H} \alpha_j p^j$ where $\alpha_i = 0$ or $1$ and $\alpha_j = 0, 1, 2 \cdots, L-1$. More specifically, if $p$ contains $w_i$, $i \leq z$, then $\alpha_i = 1$; otherwise, $\alpha_i = 0$.

For the second statement, because the activation status is determined by all the $o_{i_l}^l(x)$, we just need to prove the sign of $o_{i_l}^l(x)$ is determined by the value of basis path vector $v$. For each hidden node $O_{i_l}^l$, there exist only one basis path which passes it and only contain skeleton weights (We call the basis path which contains only skeleton weights *all-basis path*). We denote the value of all-basis path which passes $O_{i_l}^l$ as $v_{p^a}(O_{i_l}^l) = w_1(O_{i_l}^l) \cdot \prod_{j=2}^L w_j(O_{i_l}^l)$, where $w_j(O_{i_l}^l)$ denotes the skeleton weight of $p^a(O_{i_l}^l)$ at the $j$-th layer which is also an skeleton outgoing weight for one hidden node. We will prove $o_{i_l}^l(x)$ can be calculated as

$$o_{i_l}^l(x) = \frac{1}{\prod_{j=l+1}^L w_j(O_{i_l}^l)} \cdot F_{i_l}^l(v; x), \tag{11}$$

where $F_{i_l}^l(v; x)$ is a function which is determined by $v$ and the input $x$. If Eqn(11) is satisfied, the sign of $o_{i_l}^l(x)$ can be determined as following:

$$sgn(o_{i_l}^l(x)) = sgn(w_{l+1}(O_{i_l}^l)) \cdots sgn(w_L(O_{i_l}^l)) \cdot sgn(F_{i_l}^l(v; x)). \tag{12}$$

Next we prove Eqn(11) by induction.

(1) For $l = 1$,

$$o_{i_1}^1(x) = \sum_{i_0=1}^{h_0} w_1(i_0, i_1)) \cdot x_{i_0} = \sum_{i_0=1}^{h_0} \frac{v_{p^s}(w_1(i_0, i_1))}{\prod_{j=2}^L w_j(O_{i_1}^1)} \cdot x_{i_0} = \frac{1}{\prod_{j=2}^L w_j(O_{i_1}^1)} \sum_{i_0=1}^{h_0} p^s(w_1(i_0, i_1))) \cdot x_{i_0}, \tag{13}$$

where $v_{p^s}(w_1(i_0, i_1)))$ is the value of basis path which contains $w_1(i_0, i_1)$ and $w_j(O_{i_1}^1)$ is the outgoing skeleton weight (free skeleton weight) of $O_{i_1}^1$. It means that Eqn(11) is satisfied with $F_{i_1}^1(v; x) = \sum_{i_0=1}^{h_0} v_{p^s}(w_1(i_0, i_1))) \cdot x_{i_0}$.

(2) For $l > 1$, suppose that $o_{i_{l-1}}^{l-1}(x) = \frac{1}{\prod_{j=l}^L w_j(O_{i_{l-1}}^{l-1})} \cdot F_{i_{l-1}}^{l-1}(v; x)$,

$$o_{i_l}^l(x) = \sum_{i_{l-1}=1}^{h_{l-1}} w_l(i_{l-1}, i_l) \cdot o_{i_{l-1}}^{l-1} \tag{14}$$

$$= \sum_{i_{l-1}=1}^{h_{l-1}} \frac{v_{p^s}(w_l(i_{l-1}, i_l))}{\prod_{j=l+1}^L w_j(O_{i_l}^l)} \cdot o_{i_{l-1}}^{l-1} \tag{15}$$

$$= \sum_{i_{l-1}=1}^{h_{l-1}} \frac{v_{p^s}(w_l(i_{l-1}, i_l))}{\prod_{j=l+1}^L w_j(O_{i_l}^l) \cdot w_1(O_{i_{l-1}}^{l-1}) \cdot \prod_{j=1}^{l-1} w_j(O_{i_{l-1}}^{l-1})} \cdot \frac{1}{\prod_{j=l}^L w_j(O_{i_{l-1}}^{l-1})} \cdot F_{i_{l-1}}^{l-1}(v; x) \tag{16}$$

$$= \frac{1}{\prod_{j=l+1}^L w_j(O_{i_l}^l)} \sum_{i_{l-1}=1}^{h_{l-1}} \frac{v_{p^s}(w_l(i_{l-1}, i_l))}{w_1(O_{i_{l-1}}^{l-1}) \cdot \prod_{j=1}^{l-1} w_j(O_{i_{l-1}}^{l-1})} \cdot \frac{1}{\prod_{j=l}^L w_j(O_{i_{l-1}}^{l-1})} \cdot F_{i_{l-1}}^{l-1}(v; x) \tag{17}$$

$$= \frac{1}{\prod_{j=l+1}^L w_j(O_{i_l}^l)} \sum_{i_{l-1}=1}^{h_{l-1}} \frac{v_{p^s}(w_l(i_{l-1}, i_l))}{v_{p^a}(O_{i_{l-1}}^{l-1})} \cdot F_{i_{l-1}}^{l-1}(v; x) \tag{18}$$

$$= \frac{1}{\prod_{j=l+1}^L w_j(O_{i_l}^l)} \cdot F_{i_l}^l(v; x). \tag{19}$$

Thus we have finished the proof the second statement.

## 9.3 PROOF OF COROLLARY 3.8

**Corollary 3.8** *The dimension of $\mathcal{G}$-space is $m - H$, where $m$ is the number of weights and $H$ is the total number of hidden nodes for MLP or the total number of feature maps for CNN.*

*Proof:* The dimension of a mathematical space (or object) means that the minimum number of coordinates needed to specify any point within it. When the signs of free variables are fixed, the number of the variables which are used to represent the output of ReLU neural networks is $m - H$ according to Theorem 3.4. Thus the dimension of $\mathcal{G}$-space is $m - H$.

## 9.4 PROOF OF THEOREM 4.1

**Theorem 4.1:** *Suppose that there are two different projections $\mathcal{T}_1$ and $\mathcal{T}_2$ that project the path-ratio to weight-ratio. If the projection will not change the signs of free skeleton weights, the values of basis paths will keep the same at every iteration for two $\mathcal{G}$-SGD processes that are initialized with the same values of basis paths and use different projections $\mathcal{T}_1$ and $\mathcal{T}_2$ in WA method respectively.*

*Proof:* In fact, the gradients of basis paths only depends on the values of basis paths and are independent with how the weights distribute.

Specifically, suppose that the two training processes start from the same initial point, i.e., $v^0_{(1)} = v^0_{(2)}$. Because the values of paths and the activation status can be calculated using values of basis paths when the signs of free skeleton weights are fixed, the loss functions can be represented using the values of basis paths. Then the neural network with equal values of basis paths will produce the same gradient of basis paths.

After one step of SGD, we have $v^1_{(1)} = v^1_{(2)}$. Then all the path ratios are also the same, i.e., $R^0_{(1)}(p^i) = R^0_{(2)}(p^i)$ for every basis path $p^i$. Then we use $\mathcal{T}_1$ and $\mathcal{T}_2$ to project the path ratios to weight ratios. Although $\mathcal{T}_1(R^0_{(1)}(p^i)) \neq \mathcal{T}_2(R^0_{(2)}(p^i))$, the two networks are still have equal values of basis paths after the projection, which means they are still in the same equivalent class. Again, the neural network with equal values of basis paths will produce the same gradient of basis paths. So the values of basis paths will always keep the same during the $\mathcal{G}$-SGD process.

## 10 APPENDIX INFORMATION OF THE $\mathcal{G}$-SGD ALGORITHM

### 10.1 UPDATE RULE AND TIME COMPLEXITY OF $\mathcal{G}$-SGD

Suppose that $p^i$ with $i = 1, \cdots, z$ is the basis path containing one non-skeleton weight (denoted as $w_i$), and $p^j$ with $j = z + 1, \cdots, m - H$ is the basis path containing skeleton weights only, $w_j$ is its skeleton weights at layer 1, and $w_k$ is the free skeleton weights that are not updated.

First, according to the ICR Method, we can get the update rule of value of skeleton paths as below,

$$v^{t+1}_i = v^t_i - \eta_t \frac{\delta^t_{w_i} \cdot w^t_i}{v^t_i} \tag{20}$$

$$v^{t+1}_j = v^t_j - \eta_t \frac{w^t_j \cdot (\delta^t_{w_j} - \sum_{p^i:w_j} \delta^t_{v_i} \cdot \frac{v^t_i}{w^t_j})}{v^t_j} \tag{21}$$

Combined with the weight allocation method, we can get the update rule of $\mathcal{G}$-SGD as follows:

$$w^{t+1}_j = w^t_j - \eta_t \cdot \frac{\delta^t_{w_j} \cdot (w^t_j)^2 - w^t_j \cdot \sum_{p^i:w_j} \delta^t_{w^t_i} \cdot w^t_i}{(v^t_j)^2} \tag{22}$$

$$w^{t+1}_i = \frac{w^t_i - \eta_t \cdot \frac{\delta^t_{w_i} \cdot (w^t_i)^2}{(v^t_i)^2}}{r^t(w_j : p^i)}, \tag{23}$$

$$w^{t+1}_k = w^t_k, w_k \neq w_i, w_j \tag{24}$$

where $r^t(w_j : p^i)$ is ratio of the skeleton weight $w_j$ at layer 1 which is contained in basis path $p^i$.

If the free skeleton weights $w_k$ are initialized as 1, then $v^t_j = w^t_j$. Thus we can first calculate the path-ratio for all-basis paths:

$$R^t(p^j) = \frac{v^{t+1}_j}{v^t_j} = 1 - \eta_t \frac{w^t_j \cdot (\delta^t_{w_j} - \sum_{p^i:w_j} \delta^t_{v_i} \cdot \frac{v^t_i}{w^t_j})}{(v^t_j)^2} = 1 - \eta_t \frac{\delta^t_{w_j} - \sum_{p^i:w_j} \delta^t_{v_i} \cdot \frac{v^t_i}{w^t_j}}{w^t_j}. \tag{25}$$

According to the WA method, $r^t(w_j : p^i) = R^t(p^j)$. Then the update rule for different kinds of weights can be classified into the following three types

$$w^{t+1}_j = w^t_j \cdot R^t(p^j) \tag{26}$$

$$w^{t+1}_i = \frac{w^t_i - \eta_t \cdot \frac{\delta^t_{w_i} \cdot (w^t_i)^2}{(v^t_i)^2}}{r^t(w_j : p^i)}, \tag{27}$$

$$w^{t+1}_k = w^t_k, w_k \neq w_i, w_j \tag{28}$$

where $r^t(w_j : p^i)$ is ratio of the skeleton weight $w_j$ at layer 1 which is contained in basis path $p^i$, $w_j$ denotes the skeleton weight at layer 1, $w_i$ denotes the non-skeleton weight and $w_k$ denotes the free skeleton weights (the skeleton weights that not in layer 1).

The forward and backward propagation take the dominating time in both mini-batch SGD and $\mathcal{G}$-SGD. To be specific, if the mini-batch size is $B$, the forward and backward propagation would both take

$BT$ time. Comparing with SGD, the extra computation is the calculation of $R^t(p^j)$, which the time complexity is independent with the mini-batch and is equal to the update step in SGD. Thus the time complexity of $\mathcal{G}$-SGD is upper bound by $\mathcal{O}((B+1)T)$. Please see the training throughput of our experiments on GPU server in Section 11.1.

## 10.2 Skeleton Method for ResNet and ICR Method for Batch Normalization

For ResNet, we implement the skeleton method to construct skeleton weights and basis paths in each residual block. Because of the skip-connection, there is an identity weight which doesn't change during the optimization. Thus, if the skip-connected weight connects node $O$, there isn't a positive scaling operator $g_{O,c} \in \mathcal{G}$ to make $w \sim g_{O,c}(w)$. So we can't construct basis paths for the whole network. The invariance of ResNet only exists in each residual block. In this sense, the equivalence of invariance is less severe than other neural network structure.

Because of the existence of the batch-normalization layers, the output of neural network with BN is $\hat{o_{ij}}^l = \frac{o^l_{ij} - \mu_j}{\sqrt{\sigma_j^2 + \epsilon}}$, where $o^l_{ij}$ means the $j$-th output in layer-$l$ which is calculated using the $i$-th sample, $\mu_j = \frac{1}{b}\sum_{i_0=1}^{b} o^l_{ij}$ is the expectation of $o_{ij}, i = 1, \cdots, b$ and $\sigma_j^2 = \frac{1}{b}\sum_{i=1}^{b}(o^l_{ij} - \mu_j)^2$ is the variance. Assume that $o^l_{ij} = w^l_j o^{l-1}_i$ and the inputs $o^{l-1}_i$ has expectation 0 and variance 1 (It can be roughly satisfied for neural networks with BN.), we have $\sigma_j^2 \approx \|w^l_j\|^2$. If we define the value of a path as $v_w(p) = \prod_{l=1}^{L} \frac{w^l(i_{l-1}, i_l)}{\|w^l_{i_l}\|}$ where $w^l_{i_l}$ denotes the incoming weight vector of node $O^l_{i_l}$. Thus the loss function of the NN with BN layers can be approximately represented as $l(v; x, y)$.

Thus inverse-chain-rule for NN with BN layer can be approximated by the following equations. If $w$ is an incoming weight of node $O$, we have

$$\frac{\partial l(w; x, y)}{\partial w^l(i_{l-1}, i_l)} \approx \sum_{i=1}^{z_1} \frac{\partial l(v; x, y)}{\partial v_i} \cdot \frac{\partial v_i}{\partial w^l(i_{l-1}, i_l)} \cdot \frac{1}{\|w^l_{i_l}\|}, \qquad (29)$$

which results in

$$\frac{\partial l(w; x, y)}{\partial w^l(i_{l-1}, i_l)} \cdot \|w^l_{i_l}\| \approx \sum_{i=1}^{z_1} \frac{\partial l(v; x, y)}{\partial v_i} \cdot \frac{\partial v_i}{\partial w^l(i_{l-1}, i_l)}. \qquad (30)$$

Then we use the above equation in the ICR methods.

## 11 Appendix Information of the Experiments

### 11.1 Training Throughput of $\mathcal{G}$-SGD

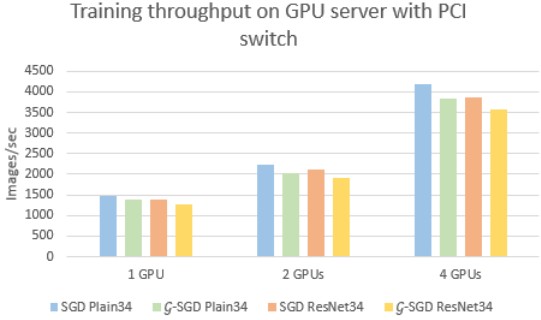

Figure 6: Training throughput on GPU server with 4 NVIDIA GTX Titan Xp GPUs and PCI switch. Mini-batch size per GPU is always 128.

We implement our $\mathcal{G}$-SGD using the *Pytorch* framework with v0.31 stable release, and conduct our experiments comparing with Pytorch built-in SGD optimizer. Our experiments are conducted on a

GPU server with 4 NVIDIA GTX Titan Xp GPUs and PCI switch. We show the training throughput (processing images per second) on CIFAR-10 dataset with SGD and $\mathcal{G}$-SGD optimizer on different network architectures. The multi-GPU training is done by Pytorch built-in multiple GPU module *torch.nn.parallel.DataParallel* based on NCCL. As shown in Figure 6, when the mini-batch size is set to 128, the training throughput of $\mathcal{G}$-SGD is slightly lower than vanilla SGD by about 7%, which indicates that our implementation of $\mathcal{G}$-SGD is indeed efficient.

## 11.2 INITIALIZATION METHOD OF SKELETON WEIGHTS

According to our analysis in the main paper, only the signs of skeleton weights matter the optimization. Thus we need to determine the signs of skeleton weights before training process. For the absolute value of skeleton weights, we can see from section 10.1 that different absolute value of skeleton weights well determine different scale of learning rate. Although our theoretical results show that the absolute value of skeleton weights can be randomly set, we choose them to be 1 for easier learning rate tuning and robustness.

In order to verify how signs of skeleton weights influence the performance. We test the performance for various combination of signs for them on image classification task (see section 5.2). Results shows that there are no differences for them. A intuitive explanation is that the selected network model is over-parameterized and the approximation ability will not be influenced by signs of skeleton weights. For simplicity, we initialize the value of skeleton weights as 1.

## 11.3 OPTIMIZING 110-LAYER RESNET WITH $\mathcal{G}$-SGD

In this section, we study the optimization performance of $\mathcal{G}$-SGD on deeper ResNet model with 110 layers. We employ the same training strategies on both model structures, which are detailed described in next subsection. As we can see in the Figure 7 and Table 5, our $\mathcal{G}$-SGD clearly outperforms SGD on both shallow and deep ResNet model. The best test accuracy on ResNet-110 are achieved with $\mathcal{G}$-SGD on both dataset, which meets the same conclusions in Section 5.1.Please note that deeper ResNet doesn't significantly outperform shallow ResNet on CIFAR-100 dataset, which also be observed in the original ResNet paper He et al. (2016a) and the authors propose a new ResNet architecture to solve this problem in He et al. (2016b).

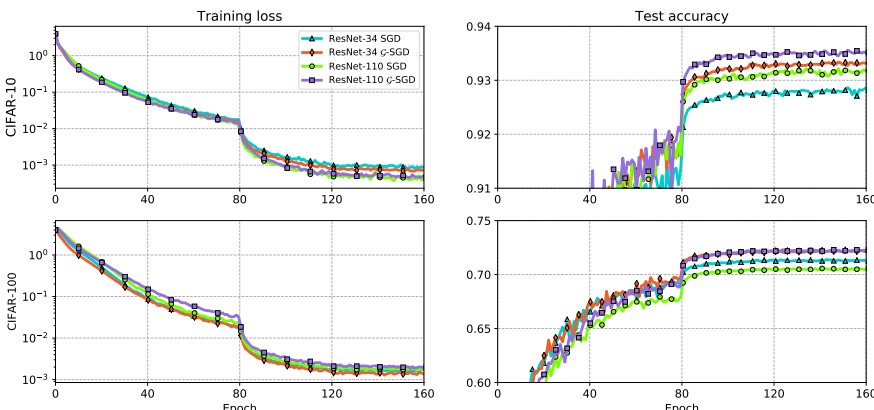

Figure 7: Training loss and test accuracy on 110-layer ResNet network w.r.t. the number of effective passes on CIFAR-10 and CIFAR-100 dataset.

## 11.4 DETAILED TRAINING STRATEGIES IN SECTION 5.1 AND 11.2

In previous experiment section, we extend our $\mathcal{G}$-SGD to deep convolutional networks. CIFAR-10 and CIFAR-100 have been used in the experiment. We apply random crop to the input image by size of 32 with padding 4, and normalize each pixel value to [0,1]. We then apply random horizontal flipping to the image. The mini-batch size of 128 is used in this experiment. The training is conducted for 64k iterations. We follow the learning rate schedule strategy in the original paper

Table 5: Classification error rate (%) on image classification task.

| | | C10 | C100 |
|---|---|---|---|
| ResNet-34 | SGD | 7.13 ($\pm$0.22) | 28.60($\pm$0.51) |
| | $\mathcal{G}$-SGD | **6.66** ($\pm$0.13) | **27.74** ($\pm$0.24) |
| ResNet-110 | SGD | 6.83 ($\pm$0.25) | 29.44($\pm$0.66) |
| | $\mathcal{G}$-SGD | **6.49** ($\pm$0.06) | **27.74** ($\pm$0.36) |

(He et al., 2016a), specifically, the initial learning rates of vanilla SGD and $\mathcal{G}$-SGD are set to 1.0 and then divided by 10 after 32k and 48k iterations. The ResNet implementation can be found in https://github.com/pytorch/vision/ and the models are initialized by the default methods in PyTorch.

## 11.5 THE COMBINATION OF $\mathcal{G}$-SGD AND REGULARIZATION

The optimization algorithms achieve better generalization performance on test dataset by combining with proper regularization methods. In the previous experiments, we focus on the difference performance of optimization algorithms. In this section, we conduct the experiments to investigate the combination of $\mathcal{G}$-SGD and regularization. In weight space, weight norm is widely used as regularization for ReLU networks (He et al., 2016a; Huang et al., 2017). Recently, (Zheng et al., 2018) propose the basis path norm in $\mathcal{G}$-space. In this section, we reproduce the experiments in (He et al., 2016a; Zheng et al., 2018) on SGD regularized by weight norm (SGD+WD) and $\mathcal{G}$-SGD regularized by basis path norm ($\mathcal{G}$-SGD+BPR), and extend them on CIFAR-100 dataset. The learning rate of 1.0 is widely used to train ResNet model and its variants on CIFAR dataset, hence we employ it in our experiment as well. We do a wide range grid search for the hyper-parameter $\lambda$ for weight decay and basis path regularization from $\{0.1, 0.2, 0.5\} \times 10^{-\alpha}$, where $\alpha \in \{3, 4, 5, 6\}$, and report the best performance based on the CIFAR-10 validation set. We use the same hyper-parameter on CIFAR-100 dataset.

## 11.6 DETAILED TRAINING STRATEGIES IN SECTION 5.2

In this section, our aim is to verify the influence of invariance to optimization in weight space. We train several 2-hidden-layer MLP models with different invariant ratio (i.e. $H/m$) on Fasion dataset. The original size of input image is $28 \times 28$. We normliazed the input data, and to reduce the dimensions of input feature, we downsample the image to $7 \times 7$ by using average pooling. The network structue is followed by [49:$h$:$h$:10] where $h$ is the number of hidden nodes in each layer. The detailed model properties are shown in table 6. All models are initialized by (He et al., 2015) except the skeleton weights which is mentioned in Section 4 without explicit note. We use the learning rate of 0.01 and mini-batch size of 64 for vanilla SGD and $\mathcal{G}$-SGD, and train each model for 100 epochs.

Table 6: Network information in Section 5.2.

| #hidden nodes | 8 | 16 | 32 | 64 | 128 | 256 | 512 | 1024 |
|---|---|---|---|---|---|---|---|---|
| #weights | 536 | 1200 | 2912 | 7872 | 23936 | 80640 | 292352 | 1108992 |
| $H$ | 16 | 32 | 64 | 128 | 256 | 512 | 1024 | 2048 |
| invariant ratio | $1.49 \times 10^{-2}$ | $1.33 \times 10^{-2}$ | $1.10 \times 10^{-2}$ | $8.13 \times 10^{-3}$ | $5.35 \times 10^{-3}$ | $3.17 \times 10^{-3}$ | $1.75 \times 10^{-3}$ | $9.23 \times 10^{-4}$ |

As shown in Figure 8 and 9, the $\mathcal{G}$-SGD achieves clearly better performance than SGD with combining regularization method on all models and all datasets. To be specific, the test accuracy of 94.29% is gained by ResNet-34 trained by SGD and weight decay on CIFAR-10 dataset (the number reported in (He et al., 2016a) is 91.25%), while $\mathcal{G}$-SGD with basis path regularization improves the test accuracy to 94.67%. On CIFAR-100 dataset, the test accuracy of ResNet-34 trained by SGD and weight decay is 74.39% (the ResNet-34 result on CIFAR-100 hasn't been reported in He et al. (2016a), a result of ResNet-110 with similar training strategy on this dataset is reported in Zagoruyko & Komodakis (2016) which is 74.84%), while $\mathcal{G}$-SGD with basis path regularization improves the test accuracy to 75.20%. The experimental results verify our analysis again that optimization in $\mathcal{G}$-space is a better choice.

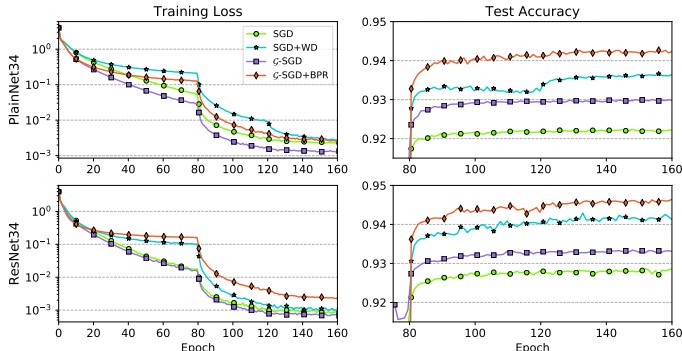

Figure 8: Training loss and test accuracy w.r.t. the number of effective passes on CIFAR-10 dataset.

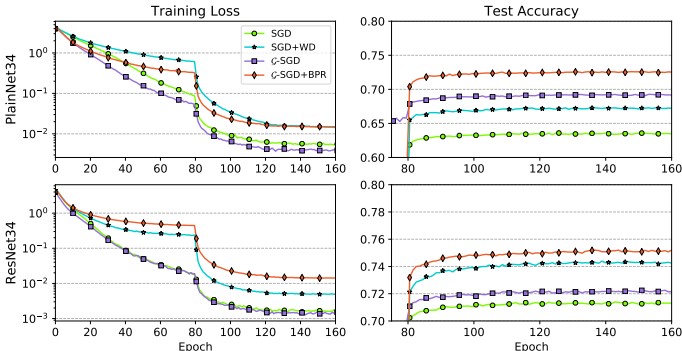

Figure 9: Training loss and test accuracy w.r.t. the number of effective passes on CIFAR-100 dataset.

## 11.7 DETAILED TRAINING STRATEGIES IN SECTION 5.3

Path-SGD (Neyshabur et al., 2015) also notice the positive scale invariance in neural networks with linear or ReLU activation. Instead of optimizing the loss function in $\mathcal{G}$-space, they use path norm as regularizer to the gradient in weight space. Meanwhile, the dependency among all paths hasn't been noted, which leads to the computation overhead of the gradient of path norm is very high. In section 5.3, we extend the experiments in (Neyshabur et al., 2015) to $\mathcal{G}$-SGD on MNIST and CIFAR-10 dataset. A 5-hidden-layer MLP model is used in this experiment with 64 units in each layer. We do grid search for the learning rate of each algorithm from $1.0 \times 10^{-\alpha}$, where $\alpha$ is an integer between 0 to 10. We report the best result for each algorithm. The mini-batch size of 64 is used, and the input images of gray scale are normalized to $[0, 1]$.

