# OpenReview forum: "G-SGD: Optimizing ReLU Neural Networks in its Positively Scale-Invariant Space"
_ICLR.cc/2019/Conference_

### Official Review · AnonReviewer3 · 2018-11-02
**interesting idea, but not convincing exps**

**Rating:** 7
**Confidence:** 4

**Review:**

This paper proposed a new training algorithm, G-SGD, by exploring the positively scale-invariant space for relu neural networks. The basic idea is to identify the basis paths in the path graph, and convert the weight update in SGD to the weight rescaling in G-SGD. My major concerns are as follows:

1. Empirical significance of G-SGD: While the idea of exploring the structures of relu neural networks for training based on group theory on graphs is interesting, I do not see significant improvement over SGD. The accuracy differences in Table 1 are marginal, training/testing behaviors in Fig. 3 are very similar, and more importantly there is no evidence to support the claims "with little extra cost" in the abstract/Sec. 4.3 in terms of computation. Therefore, I do not see the truly contribution of the proposed method.

PS: After reading the revision, I am happy to see the results on computational time that support the authors' claim. However, I still have doubts on the significance of the improvement on CIFAR10 and CIFAR100, because the performance is heavily dependent on network architectures. In my experience, using resnet101 it can easily achieve >96% accuracy. So can you achieve better than this using G-SGD? The training and testing behaviors on both datasets somehow show improvement over SGD, which I take it more importantly than just those numbers. Therefore, I am glad to raise my score.

2. In Alg. 3 I do not quite understand how to apply step 3 to step 4. The connection needs more explanation.

---

> ### Author Response · Authors · 2018-11-15
> **Thanks for your reviewing and suggestions**
>
> Thank you for the helpful feedback.
>
> 1. Q: Significance of experiments in Table 1.Similar training/testing behaviors in Figure 3.
> The performance gain by G-SGD in Table 1 is not marginal. For example, since it can eliminate the influence of positive scaling invariance across all layers of PlainNet, G-SGD can averagely improve the accuracy number by 0.8 and 5.7 on CIFAR-10 and CIFAR-100 respectively, over 5 independent runs. Moreover, Plain-34 trained by G-SGD achieves even better accuracy than ResNet-34 trained by SGD on CIFAR-10, which shows the influence of invariance on optimization in weight space as well.
> In our experiments, we employ the training strategies in the original ResNet paper [1] (which is designed for SGD) for both SGD and our G-SGD. This is the reason why the training/testing behaviors are similar. In the future, we will design training strategies for optimization in G space, which will further enhance its performance.
>
> 2. Q: Evidence of low computational cost.
> By the update rule of G-SGD in Section 11.1 (i.e., Eqn (43) and Eqn(46)), the extra computation is the calculation of $R^t(p^j)$. It is clear that the computation overhead of our G-SGD is upper bounded by $(B+1)T$ (B is the batch size and T is the time of processing one sample, and the cost of SGD is BT). Considering that the computation cost of original SGD is $BT$, G-SGD has “little extra cost”. We will add the complexity analysis and the figure of training throughput on our GPU server in the appendix. Please see the next version of our paper.
>
> 3. Q: Connection between step 3 and 4 in Alg.3.
> We assume your comments are about Alg 1, since there is no Alg. 3 in our paper. Please let us know if not. In step 3, we calculate the gradient with respect to basis path by solving Eqn (3) in Inverse-Chain-Rule method. Then, we update basis paths according to SGD accordingly. After that, we can compute the update ratio for basis path. In step 4, we allocate the update of basis path to the update ratio of weight.
>
> Thank you for the valuable suggestion. We will add the connection between step 3 and step 4 in the next version.
>
> [1] He K, Zhang X, Ren S, et al. Deep residual learning for image recognition[C]//Proceedings of the IEEE conference on computer vision and pattern recognition. 2016: 770-778.

---

> ### Author Response · Authors · 2018-11-22
> **Thanks for raising the score**
>
> Thank you for raising the score. We’re glad to see that our response has addressed your concerns.
>
> We also want to investigate how G-SGD performs on deeper ResNet (such as 101 layers or more).  In our mind, deeper neural networks may benefit more from the optimization in G-space.  We are conducting the experiments and try our best to add the results before the rebuttal deadline.
>
> One of our concern is that, to the best of our knowledge, with the traditional or similar training strategies, ResNet-101 can easily achieve about 93%+ accuracy on the CIFAR-10 dataset, but a little bit far from >96%. For example, 93.75% and 93.89% are reported by these two PyTorch-based implementations(https://github.com/kuangliu/pytorch-cifar and https://github.com/bearpaw/pytorch-classification), respectively. As for comparison, in our implementation, the test accuracy of 94.29% is gained by ResNet-34 trained by SGD and weight decay. Therefore, it will be helpful to tell us if there is any implementation or training strategy that can train ResNet-101  with 96% test accuracy.

---

> ### Author Response · Authors · 2018-12-18
> **Add experiments on deeper ResNet**
>
> We have finished the experiments on 110-layer ResNet (He, et. al. 2016) using the same training strategies with Table1 in the paper. The test error rates are shown below:
> --------------------------------------------------------------------------
>                      CIFAR-10                 CIFAR-100
> SGD           6.83% (±0.25)         29.44%(± 0.66)
> G-SGD       6.49%(±0.06)          27.74%(±0.36)
> --------------------------------------------------------------------------
> It shows that G-SGD clearly outperforms SGD on each dataset. The best test accuracies are achieved by  G-SGD on both datasets, which indicates that G-SGD indeed helps the optimization of deep ResNet model. We have added the results in the draft and will update the paper in the next version.

---

### Official Review · AnonReviewer2 · 2018-11-02
**SGD is recast in positively scale-invariant space, showing improvements over training in weight space with low computational overhead.**

**Rating:** 7
**Confidence:** 3

**Review:**

Summary:
In prior deep learning papers it has been observed that ReLU networks are positively scale invariant due to the non-negative homogeneity property of the max(0, x) function. The present paper proposes to change the optimization procedure so that it is done in a space where this invariance is preserved (in contrast to the weight space, where it is not). To do so, the authors define the group G of positive scaling operators, and note that the "value of path" (product of weights along the path) is G-invariant and together with "activation status of paths" allows for the definition of an equivalence class. They then build a G-space which has fewer dimensions than the weight space, and proposed g-SGD to optimize the network in this space. In g-SGD gradients are computed normally, then projected to G-space via a sparse matrix in order to update the values of paths. The weights are then updated based on a "weight allocation method" that involves the inverse projection.

The authors conduct experiments on CIFAR-10 and -100 with a ResNet-34 and a similarly deep variant of VGG, showing significant benefits from G-SGD training in all cases. They also evaluate the performance of a simple MLP on Fashion-MNIST as a function of the invariant ratio H/m.

Comments:
The paper is organized well (with technical details of the proofs delegated to the appendix), and discusses the differences in comparison prior work. While evaluations on large-scale datasets would be helpful here, the present experiments suggest that optimization in G-space indeed consistently improves results, so the proposed method seems promising.

For completeness, it would be great to include Path-SGD results for the CIFAR experiments in Table 1, together with runtime information to highlight the benefits of the g-SGD algorithm and provide experimental proof that the computational overhead is indeed low.

If the authors are hoping for a wider adoption of the method, it would be helpful for the community to have the g-SGD code released within one of the standard deep learning frameworks.

Questions:
- Does it matter which weights are chosen as "free skeleton weights"? If these weights never get updated in the optimization procedure, could you please comment on the intuitive interpretation of the necessity of their presence?
- The text states that the computational overhead of the gradient of path norm is "very high". The Path-SGD paper proposes a method costing (B + 1)T, where the overhead an be small for large batches. It would be good to clarify this a bit in the present text.
- Is the advantage of g-SGD over SGD expected to be proportional to the invariant ratio for CNNs as well?

---

> ### Author Response · Authors · 2018-11-15
> **Thanks for your reviewing and suggestions**
>
> Thank you for the valuable comments which are addressed as follows.
>
> 1. Q: Path-SGD results for the CIFAR experiments?
> It is unaffordable to compute gradients  with respect to all the paths, thus the authors in [1] take coordinate-independent approximation in Path-SGD (Eqn(7) in [1] can be referred). The authors design Path-SGD only for MLP[1] and RNN[2],  and  the Path-SGD for CNN is non-trivial. Therefore, the result of Path-SGD on ResNet and PlainNet is not provided in Table 1. We do compare with Path-SGD for MLP case in Figure 5.
>
> 2. Q: Experimental proof of computational overhead?
> Thanks for your advice. We will add the figure of training throughput on our GPU server in the appendix. Please see the next version of our paper.
>
> 3. Q: Code Release within one of the standard deep learning frameworks.
> We plan to open source the codebase of our implemented G-SGD, not only for reproducing our experimental results, but also as a toolkit which can be freely used for the community.
>
> 4. Q: The chosen of “free skeleton weights”? The necessity of their presence?
> Different selections of free skeleton weights are mathematically equivalent and all the theoretical results will preserve, e.g., the activation status is a function of basis path if the signs of free skeleton weights are fixed.
> Actually we can project the gradient with respect to basis path back to any weights on that path, but according to weight-allocation method, we want to reduce the number of update executions to reduce the time complexity. Hence, we choose one skeleton weight to update on all-basis path (which is only composed by skeleton weights) and others are selected to be free skeleton weights.
> Although there is no update on those “free skeleton weights”,  they are necessary because the “value of path” is composed by all weights on one path. We can delete a weight only when it always takes value “0”.  Free skeleton weights are not initialized as “0”, and they indeed play a role in calculating the value of path and also in feedforward process in G-SGD.
>
> 5. Q: High computational overhead of the gradient of path norm?
> The gradient of path regularizer is hard to calculate exactly and a coordinate-independent approximation is made in Path-SGD which  costs (B + 1)T (B is the batch size and T is the time of processing one sample, and the cost of SGD is BT). The computation cost of our G-SGD is also (B+1)T (Please refer to Eqn(43)-Eqn(46)), which is comparable with the approximation of Path-SGD. We will clarify this comparison in the next version.
>
> 6. Q: Is the advantage of G-SGD over SGD expected to be proportional to the invariant ratio for CNNs as well?
> Yes, we have similar observations for CNN.
>
> [1] Neyshabur. B, et al. Path-sgd: Path-normalized optimization in deep neural networks. NIPS 2015.
> [2] Neyshabur. B, et al. Path-normalized optimization of recurrent neural networks with relu activations. NIPS 2016.

---

### Official Review · AnonReviewer1 · 2018-11-02
**good paper**

**Rating:** 7
**Confidence:** 2

**Review:**

The paper proposes SGD for ReLU networks. The authors focuses on positive scale invariance of ReLU which can not be incorporated by naive SGD. To overcome this issue, a positively scale-invariant space is first introduced. The authors show the SGD procedure in that space, which is based on three component techniques: skeleton method, inverse-chain rule, and weight allocation.

The basic idea, directly optimizing weight in scale invariant space, is reasonable and would be novel, and experiments verify the claim. Readability might be low slightly.

Analysis about invariance group (e.g., theorem 3.6) is interesting and informative.

Combining with other optimization algorithms (other than simple SGD) method would be valuable.

---

> ### Author Response · Authors · 2018-11-15
> **Good suggestions**
>
> Thank you for viewing the idea novel and the analysis informative.
>
> New optimization algorithm in G-Space is an interesting topic. In this work, we mainly focus on introducing G-Space through the PSI property of ReLU Neural Networks and investigate the most popular SGD algorithm in G-Space. We will further investigate new optimization algorithms in G-Space.

---

### Public Comment · (anonymous) · 2018-11-07
**BN networks are scale-invariant, therefore positively scale-invariant?**

If I understand right, BN (batch norm) networks are invariant to the scaling of the weights. Does that mean SGD is enough for BN networks? If so, do we still need G-SGD for BN networks?

---

> ### Author Response · Authors · 2018-11-08
> **G-SGD can solve the scaling-invariant problem for BN networks**
>
> It’s a good question.
>
> First, you may misunderstand the definition of positively scale-invariant in our paper. The PSI property defined in our paper is: “the output of the ReLU network is invariant if the incoming weights of a hidden node are multiplied by a positive constant “c” and the outgoing weights are divided by “c” at the same time. ”  However, BN networks are only invariant to the scaling of the incoming weights, which is different from the PSI property defined in our paper.
>
> Second, we still need G-SGD for BN networks. 1) From theoretical view, BN networks still encounter the problem that equivalent BN networks with different scaling weights generate different gradients. Consider that the incoming weights are multiplied by a constant “c”, the new parameterized network generates the same output as the previous one. However, their gradients on weights are not the same using SGD, which may hurt the optimization and is unstable about the unbalanced initialization. Thus, we choose to optimize the invariant variables – the value of basis path (the multiplication of normalized weights which is normalized by its incoming weight norm (refer to section 11.2)) for this kind of scaling operators brought by BN. Then their gradients on basis paths are the same using G-SGD. 2) Experiments also show that G-SGD helps to improve the performance with BN networks, especially on PlainNet (Figure 1). It indicates that G-SGD can help to solve the scaling invariant property brought by BN.

---

### Author Response · Authors · 2018-11-15
**The latest paper version**

Dear reviewers,

We modified our paper in that:

1) As for the concern of computation cost, please refer to the time complexity analysis (section 11.1) and training throughput (section 12.1).

2) We add step 4 and step 5 in Alg.1 to connect step 3 and step 4.

3) We fix some typos especially the notations to make the paper more readable.

We also updated our initial response to all of you, for the sake of clearer clarifications and looping in the latest manuscript changes. We hope all these can make our paper more comprehensive and remove your corresponding concerns. Thanks!

---

### Meta-Review · Area_Chair1 · 2018-12-12

**Confidence:** 3
**Recommendation:** Accept (Poster)

**Metareview:**

This paper proposes a new optimization method for ReLU networks that optimizes in a scale-invariant vector space in the hopes of facilitating learning. The proposed method is novel and is validated by some experiments on CIFAR-10 and CIFAR-100. The reviewers find the analysis of the invariance group informative but have raised questions about the computational cost of the method. These concerns were addressed by the authors in the revision. The method could be of practical interest to the community and so acceptance is recommended.